# Influence of Bonding Area on Dynamic Failure Behavior of Notched Reinforced Concrete Beams

**DOI:** 10.3390/ma16020507

**Published:** 2023-01-04

**Authors:** Min Song, Zhiyong Wang, Jie Zhang, Zhihua Wang

**Affiliations:** 1College of Mechanical and Vehicle Engineering, Taiyuan University of Technology, Taiyuan 030024, China; 2Institute of Applied Mechanics, Taiyuan University of Technology, Taiyuan 030024, China

**Keywords:** reinforced concrete beam, energy dissipation, bonding area, impact velocity, finite element analysis

## Abstract

To study the effect of the bonding area on the dynamic failure process of a reinforced concrete beam with the same reinforcement ratio, the experimental and numerical researches on the impact response of notched reinforced concrete beams in the low-velocity regime (≤2.5 m/s) are presented. The tests are carried out with a drop hammer impact testing machine and then the structural responses under different impact velocities are analyzed. Additionally, the dynamic three-point bending simulation for specimens with different bonding areas, but the same reinforcement ratio is conducted. In this numerical model, the parameters of a cohesive model verified from a steel bar pullout test are applied to the bonding layer to simulate the bond-slip behavior of steel bars. Then, the energy dissipation for each component (e.g., concrete, a steel bar, and the bonding layer) are compared and discussed. The dynamic experimental results suggest that the energy absorbed during the impact process increases with the growth of the impact velocity, while the effect of the impact velocity on the reaction force can be ignored. The numerical results indicate that the failure pattern changes from a bending failure to shear failure with the increase in the bonding area and impact velocity. With the growth of the bonding area, the steel bars reach the plastic stage easily and the internal energy dissipation of the bonding layer decreases, which protects the bonding effect between the steel bar and concrete effectively.

## 1. Introduction

During the service life of reinforced concrete structures, they may be subjected to various types of dynamic loading in the form of blasts or impacts [1,2,3]. The micro-cracks in a reinforced concrete beam are gradually derived and expanded under the effect of an external load and the change in temperature, which results in the concrete cracking [4,5]. Much researches on the dynamic failure behavior of completely reinforced concrete beams have been conducted, as well as the influence of the impact velocities, and the reinforcement ratio on the dynamic failure process has been discussed. Nevertheless, the impact resistance for a reinforced concrete (RC) beam with a crack is important to evaluate the structural safety, which is still unclear. In addition, the reinforcement ratio is an important parameter for reinforced concrete beams, which is the ratio of the reinforcement to the cross-sectional area, but it cannot reflect the bonding area between the steel bars and concrete. The bonding effect between concrete and steel bars is related to the failure pattern, bearing capacity, and deflection of the RC beam. Therefore, it is necessary to study the effects of the reinforcement bonding area and impact velocity on the dynamic failure process of notched reinforced concrete beams.

The failure behaviors of complete reinforced concrete beams under varying impact velocities have been investigated by many experimental and numerical studies [6,7]. Drop-weight impact tests are considered to be an effective method to investigate the impact resistance [8]. Considering the inertial effect during the impact process, the load between the hammer and specimen is not the true bending load on the specimen [9,10,11]. One of the popular methods of accounting for inertial forces experimentally is measuring the specimen accelerations using accelerometers [8] or Digital Image Correction (DIC) technology [12,13]. While this works well for simple homogeneous specimens such as plain concrete, steel fiber reinforced concrete and other concrete specimens can be taken as homogeneous materials. Some researchers [14] took the sum of the reaction forces between the specimen and supports as the true bending load, which was applied to evaluate the impact resistance of the beams. Rossi [15] carried out the impact test of a reinforced concrete beam and Glass Fiber Reinforced Polymer (GFRP) reinforced beams and found that the reaction force of a reinforced concrete beam under an impact load can reflect the section bending moment of a beam indirectly. Fu [16] conducted the three-point bending tests for reinforced concrete beams under high impact velocities with the drop hammer machine and discussed the dynamic failure process by comparing the impact force, reaction force, and inertial force. With the help of a high-speed camera and DIC technology, the crack paths were visualized and then the transformation of the failure pattern was discussed.

Except for the dynamic bearing capacity, the energy absorbed during the dynamic failure process is also important to evaluate the impact resistance of reinforced concrete beams. Actually, part of the impact energy is used to the failure beams, yet the other part is applied to accelerate the specimen. In order to evaluate the absorbed energy for destroying the beam, the equivalent static method is proposed [17]. After comparing the simplified analytical and experimental results from drop hammer impact three-point bending tests, the effectiveness of this method was verified by El-Hadidy [18]. Zhang [9] and Ruiz [19] conducted the three-point bending tests for notched fiber reinforced concrete beams and high strength concrete beams with a drop hammer machine, and then discussed the effect of the impact velocity on the dynamic fracture energy. Kishi [20,21] compared the impact responses with the failure modes for RC beams without stirrups, and took the area enclosed by the reaction force-mid-span deflection curves as the energy absorbed for destroying the beam during the impact process. Soleimani et al. [22] found that beyond a certain impact velocity, the flexural load capacity of reinforcement concrete beams remained constant; the increase in the stress rate did not increase their load-carrying capacity. Jie Wei [7] studied the impact resistance for ultra-high performance concrete strengthened reinforced concrete beams with experimental and numerical methods and investigated the internal energy absorption for the RC column and Ultra-High Performance Concrete (HUPC) layer, respectively. Nevertheless, the energy dissipation for each of the components was still unclear.

Under impact loads, the effect of the reinforcement ratio on the dynamic failure behavior of reinforced concrete beams cannot be ignored. Some researchers have studied the influence of the reinforcement ratio on the impact response for reinforced concrete beams, whereas the effect of the bonding area on the dynamic failure behavior is hard to find. Fujikaka et al. [23] carried out a drop-weight impact test of reinforced concrete beams with different impact velocities and investigated the influence of impact velocities and the amount of longitudinal reinforcement on the impact responses of the specimens. The experimental results indicated that the maximum impact force and the maximum mid-span deflection increased as the impact velocity increased. In Adhikary’s study [24], three-dimensional finite element (FE) models of reinforced concrete beams have been established and verified with the experimental results, followed by a parametric study to investigate the influence of the longitudinal reinforcement ratio. The numerical results suggested that the ultimate load carrying capacity increased with the increment of the reinforcement ratios, and the empirical equations were proposed to predict the Dynamic Increase Factor (DIF) of the maximum resistance of RC beams under varying impact rates. Pham [25] found that the change in the reinforcement ratio does not affect the behavior of the RC beams during the first impact impulse, however, the reaction force, deflection, and failure pattern were affected at the later stage. After conducting the three-point bending tests for RC beams with different reinforcement ratios, it was found that higher steel ratios led to a stronger response up to the steel yielding or slipping, and the ultimate strength of the beam was proportional to the steel ratios [26]. However, these studies did not take into account the material parameters involved in the fracture process. Castel [27] considered the bond-slip properties of the reinforcement and found that the bonding effect between steel and concrete substantially influenced the response of the beam. Actually, the bonding area varies with the diameter of reinforcement when the reinforcement ratio remains constant. However, there are limited studies on the influence of the bonding area on the dynamic failure behavior of reinforced concrete beams under a low impact velocity. In order to investigate the effect of the bonding area of steel bars during the dynamic failure process, it is necessary to clarify the energy dissipation for each component (e.g., concrete, steel bars, and the bonding layer).

In this paper, the steel bar pullout simulation is established, where the cohesive elements with 0 thickness are inserted between the concrete and steel bars to simulate the bond-slip behavior for steel bars. After comparing with the pullout experimental result, the cohesive parameters are obtained. Subsequently, the finite element model for the drop hammer impact system is established and verified with the drop hammer three-point bending experimental results for notched reinforced concrete beams. Based on this numerical model, the influences of the impact velocity and the bonding area on the dynamic failure process and energy dissipation for each of the components are investigated. In addition, the dynamic three-point bending tests for notched reinforced concrete beams are carried out and the failure pattern and the structural response are discussed.

## 2. Experimental Program

### 2.1. Materials and Specimen Fabrication

Reinforced concrete beams with one steel bar were applied during the experiments. The maximum size of aggregates was 15 mm, and the mixing proportion (cement: water: coarse aggregate: sand) by weight was 1: 0.35: 2.62: 1.55. Cement with a strength grade of P.O 42.5 was applied in the preparation process. All the specimens have rectangular cross sections with a width (*b*) of 150 mm, a depth (*H*) of 150 mm, and a clear span length (*S*) of 0.6 *m*. A schematic of these specimens is presented in Figure 1. The depth (d) and width ($a_{0}$) of the notch are 45 and 4 mm, respectively. All the specimens were taken out after 28 days of curing in a standard curing room (temperature of 20 ℃ for and 98% humidity). The compression strength of concrete was determined from the average strength of three cubic specimens with dimensions of 150 × 150 × 150 mm^3^, and the average compressive strength was 67.39 MPa. Hotrolled Ribbed Bars (HRB) with a yield strength of 335 MPa ($f_{u}$) and an elasticity modulus of 2.06 × 10^5^ MPa ($E_{f}$) were used in the test, and the configurations of the bar were that it was 800 mm in length and 12 mm in diameter. As shown in Figure 1a, the center of the steel bar section is 30 mm away from the bottom of the beam (c = 30 mm). Additionally, a strain gauge (Model: BE120-3AA) was bonded at the center to monitor the deformation of the reinforcement, as illustrated in Figure 1b. To avoid damage to the strain gauges during the concrete casting process, the strain gauge was wrapped with gauze bonded by epoxy resin, as shown in Figure 1c.

### 2.2. Three-Point Bending Tests

All the impact tests were conducted with the drop hammer impact device designed and constructed in the Qinghai Provincial Key Laboratory of Energy-saving Building Materials and Engineering Safety, as shown in Figure 2. Soleimani [14], Zhang [28], and Li [29,30] conducted impact tests with the drop hammer weighing more than 100 kg to obtain the more stable test results. The weight of hammer applied in this test is 106.05 kg. The hammerhead is a circular plane with a diameter of 100 mm due to the effect of the hammerhead’s shape on the failure mode of the RC beams being very small [31]. The three drop heights (*H*) were 50, 150, and 250 mm, and the specimens were indicated as S1-50, S1-150, and S1-250. According to the $v=\sqrt{2gH}$, the corresponding impact velocities are 9.89 × 10^2^ mm/s, 1.71 × 10^3^ mm/s, and 2.21 × 10^3^ mm/s, respectively, where *g* is the acceleration of gravity (*g* = 9.8 m/s^2^). In this paper, the maximum impact energy is just enough to make the crack run through the specimen. Before starting the impact test, two steel supports were fasted on the rigid base symmetrically. As presented in Figure 2, three force sensors were attached to the hammer and the two supports to measure the impact force and reaction forces with a sampling rate of 2 MHz. Additionally, the electric signal of the impact force triggers the oscilloscopes. In addition, the speckle was prepared on the mid-span surface to obtain the displacement field with DIC technology. The impact process was captured by a high-speed camera (I-speed 716) with a sampling rate of 50 kHz, and the manual trigger mode was applied. The vertical displacement for the midpoint at the upper surface of the beam was regarded as the mid-span deflection, which was obtained by the DIC technique, as shown in Figure 2c.

## 3. Numerical Investigation

Finite element analysis was conducted to simulate the dynamic response of the reinforced concrete beams subjected to impact loading using a commercial Finite Element Model (FEM) software (Abaqus/CAE, ABAQUS 6.13, Dassault, Walker Rezon, France) [32]. In this paper, the Concrete Damage Plasticity (CDP) model [33] and elastic–plastic model are applied for the concrete and steel bars, respectively.

### 3.1. Constitutive Relationship

#### 3.1.1. Concrete and Reinforcement

Many scholars have analyzed the constitutive relationship of concrete under uniaxial tension and compression. Assuming that concrete is a kind of uniform, homogeneous, elastic, and brittle material. In this paper, the CDP model is applied to describe the nonlinear behavior of concrete in ABAQUS. This constitutive model considers the tensile/compressive damage and has been used to analyze the failure process of concrete under static/dynamic loads, cyclic loads, and monotonic loads [34,35,36,37]. Several experimental and numerical results have indicated that concrete is a strain rate sensitive material [38,39]. Considering that the strain rate effect has a stronger effect on the compressive and tensile strengths, only the strength enhancement behavior is examined, similarly to various previous studies [40,41]. A strain gauge was bonded at the notch root to measure the strain rate of the concrete [28], and the strain rates are in the range of 0.9–1.86 s^−1^. The strain effect of concrete has not been considered in this study.

The damage factors in the damage plastic model can describe the stiffness degradation and failure process of concrete. The stress–strain curves of concrete materials under uniaxial tensile and compressive loads are shown in Figure 3.

Under a uniaxial tensile load, the peak stress of concrete is $\sigma_{t0}$. As damage to the material occurred, the tensile stress–strain curve of concrete steps into the softening stage. In this process, a tensile fracture occurs. Additionally, the stiffness degradation of concrete is mainly realized by the tensile damage factor $d_{t}$. Under a uniaxial compression load, the yield stress of concrete is $\sigma_{cu}$. Then, the nonlinear growth trend for compressive stress is presented until the peak value is reached. As the compressive strain continues to increase, the post-peak softening stage for compressive stress can be found, where the stiffness degradation is achieved by the compressive damage factor $d_{c}$. Therefore, the tensile and compressive stresses during the softening phase can be calculated from the initial elastic modulus $E_{0}$, tensile/compressive damage factor ($d_{t}$/$d_{c}$), total tensile/compressive strain ($\varepsilon_{t}$/$\varepsilon_{c}$), and tensile/compressive plastic strain (${\tilde{\varepsilon }}_{t}^{pl}$/${\tilde{\varepsilon }}_{c}^{pl}$); the calculation formula is as follows:(1)$$\sigma_{t}=(1-d_{t})E_{0}(\varepsilon_{t}-{\tilde{\varepsilon }}_{t}^{pl})$$
(2)$$\sigma_{c}=(1-d_{c})E_{0}(\varepsilon_{c}-{\tilde{\varepsilon }}_{c}^{pl})$$

The uniaxial compressive stress of concrete can be obtained from Equation (2), where $D_{0}^{el}$ is the initial elastic matrix without damage. The constitutive equations of concrete material under uniaxial tensile loads or uniaxial compressive loads are input into the software in the form of $\sigma_{t}-{\tilde{\varepsilon }}_{t}^{ck}$ and $\sigma_{c}-{\tilde{\varepsilon }}_{c}^{in}$ to describe the deformation and stiffness degradation process. In the software, the compressive inelastic strain is the difference between the total compressive strain and the elastic strain without damage: ${\tilde{\varepsilon }}_{c}^{in}=\varepsilon_{c}-\varepsilon_{c0}^{el}$,$\varepsilon_{c0}^{el}=\frac{\sigma_{c}}{E_{0}}$. Where $\varepsilon_{c0}^{el}$ are $\varepsilon_{c}^{el}$ denote the undamaged elastic compressive strain and the compressive strain with damage, respectively, and ${\tilde{\varepsilon }}_{c}^{pl}$ and ${\tilde{\varepsilon }}_{c}^{in}$ denote the plastic strain and inelastic strain, respectively. The compressive damage parameters for concrete are inputted into the finite element software by means of $\sigma_{c}-{\tilde{\varepsilon }}_{c}^{in}$, and the inelastic strain can be turned into the plastic strain according to the following equation:(3)$${\tilde{\varepsilon }}_{c}^{pl}={\tilde{\varepsilon }}_{c}^{in}-(\varepsilon_{c}^{el}-\varepsilon_{c0}^{el})={\tilde{\varepsilon }}_{c}^{in}-\frac{d_{c}}{1-d_{c}}\frac{\sigma_{c}}{E_{0}}$$

According to the research results from other scholars [42], the calculation formula of the compression damage factor is Equation (4).
(4)$$d_{c}=1-\sigma_{c}E_{0}^{-1}/[\varepsilon_{c}^{pl}(1/b_{c}-1)+\sigma_{c}E_{0}^{-1}]$$
where the plastic strain is proportional to the inelastic strain:(5)$$\varepsilon_{c}^{pl}=b_{c}\varepsilon_{c}^{in}(0\;<\;b_{c}\;\text{≤}\;1)$$

The constitutive model of concrete under a uniaxial tensile load is expressed in the form of $\sigma_{t}-{\tilde{\varepsilon }}_{t}^{ck}$. The tensile fracture strain of concrete is defined as the difference between the total tensile strain and the elastic strain without damage: ${\tilde{\varepsilon }}_{t}^{ck}=\varepsilon_{t}-\varepsilon_{t0}^{el}$, $\varepsilon_{t}^{el}=\frac{\sigma_{c}}{E_{0}}$, where $\varepsilon_{t0}^{el}$ and $\varepsilon_{t}^{el}$ donate the no damaged tensile strain and the tensile strain with damage, respectively. The plastic strain and inelastic strain under a tension load are described as ${\tilde{\varepsilon }}_{t}^{pl}$ and ${\tilde{\varepsilon }}_{t}^{ck}$, respectively. The tensile damage parameters are inputted into the software by means of $\sigma_{t}-{\tilde{\varepsilon }}_{t}^{ck}$, then the inelastic strain is transformed into the plastic strain according to Equation (6).
(6)$${\tilde{\varepsilon }}_{t}^{pl}={\tilde{\varepsilon }}_{t}^{ck}-(\varepsilon_{t}^{el}-\varepsilon_{t0}^{el})={\tilde{\varepsilon }}_{t}^{ck}-\frac{d_{t}}{1-d_{t}}\frac{\sigma_{t}}{E_{0}}$$

Similar to the compressive damage parameters, the tensile damage parameters of concrete can be calculated as Equation (7).
(7)$$d_{t}=1-\sigma_{t}E_{0}^{-1}/[\varepsilon_{t}^{pl}(1/b_{t}-1)+\sigma_{t}E_{0}^{-1}]$$

In the reference [42], *b_c_* = 0.7 and *b_t_* = 0.1 were obtained by fitting the stress–strain curves of plain concrete subjected to repeated loads. The parameters of the concrete model are listed in Table 1.

In this model, a three-fold model is applied to the reinforcing bar. The test results show that [43], the elastic modulus of steel bars under a high strain rate remains unchanged, and the elastic modulus of the steel bars (HRB335) is 2.0 × 10^5^ MPa. The yield strength and ultimate strength of steel increase with the increase in the strain rate, and the constitutive equation can be expressed as Equations (8) and (9), respectively.
(8)$$\frac{f_{y,d}}{f_{y,st}}=1+\frac{D_{1}}{f_{y,st}}\ln\frac{\dot{\varepsilon }}{{\dot{\varepsilon }}_{0}}$$
(9)$$\frac{f_{u,d}}{f_{u,st}}=1+\frac{D_{2}}{f_{u,st}}\ln\frac{\dot{\varepsilon }}{{\dot{\varepsilon }}_{0}}$$
where $\dot{\varepsilon }$ and ${\dot{\varepsilon }}_{0}$ are the strain rate of a steel bar under dynamic loads and static loads, and 0.0003 s^−1^ is used for the static strain rate. Additionally, $f_{y,d}$, $f_{y,st}$,$f_{u,d}$, $f_{u,st}$, D1, and D2, are the dynamic yield strength, static yield strength, dynamic ultimate strength, static ultimate strength, and the coefficients obtained from experimental tests, respectively. Lin [43] conducted the tensile tests for steel bars under different loading rates, and then 6.38 and 6.54 are adopted for D1 and D2, respectively. Then, the stress–strain curve for the steel bar can be expressed as Equations (10)–(12).
(10)$$\sigma =E\varepsilon ,\;0<\varepsilon \leq \varepsilon_{b,d}$$
(11)$$\sigma =f_{y,d},\;\varepsilon_{b,d}<\varepsilon \leq \varepsilon_{e,d}$$
(12)$$\sigma =f_{y,d}+\frac{\varepsilon -\varepsilon_{e,d}}{\varepsilon_{u,d}-\varepsilon_{e,d}}f_{u,d},\;\varepsilon_{e,d}<\varepsilon \leq \varepsilon_{u,d}$$
where $\varepsilon_{b,d}$ and $\varepsilon_{e,d}$ dominates the corresponding strain at the beginning and ending of the yield platform. Additionally, $\varepsilon_{b,d}=f_{y,d}/E$,$\varepsilon_{e,d}=\varepsilon_{b,d}+\left(\varepsilon_{e,st}-\varepsilon_{b,st}\right)$, where $\varepsilon_{u,d}$ is the strain corresponding to the ultimate strength, and $\varepsilon_{u,d}=\varepsilon_{u,st}$. In this model, the yield strength and ultimate strength of the steel bar HRB335 at the strain rate of 2.9 s^−1^ are selected, as shown in Table 2.

#### 3.1.2. Constitutive Material Model Cohesive Layer

The cohesive model can reflect the mechanical properties, such as the strength and elastic modulus of the interface layer between the two materials by setting appropriate parameters. When the stress of the interface layer element reaches a certain strength, the bonding between the two materials starts to fail and the separation between the two materials is presented. In this numerical model, the cohesive model is used to simulate the bond-slip behavior of a steel bar. Thus far, the cohesive behavior between the steel bars and concrete can be described with different traction-displacement curves, such as trapezoidal, bilinear, exponential, and polynomial curves [44,45]. Among them, the bilinear model is widely used. In this paper, the bilinear traction-separation model is used to describe the bond behavior of steel bars, and the traction-separation curves are assumed to be the material properties, which can be obtained from the experimental results. The traction-displacement curves in normal and tangential as presented in Figure 4. The initial normal stiffness and tangential stiffness needs to be set high enough to represent the uncracked material, but not too high to cause numerical ill-conditioning [46]. The initial stiffness range is generally 10^4^–10^9^ MPa/mm [47,48]. In this paper, the initial stiffness of normal and tangential directions is 10^6^ MPa/mm, and the constitutive equations in normal and tangential directions are as follows:(13)$$t_{n}=\left\{\begin{array}{c} k_{n0}\delta (\delta <\delta_{n0})\\ t_{n0}\frac{\delta_{nf}-\delta }{\delta_{nf}-\delta_{n0}}(\delta >\delta_{n0}) \end{array}\right.$$
(14)$$t_{s}=\left\{\begin{array}{c} k_{s0}\delta (\delta \leq \delta_{s0})\\ t_{s0}\frac{\delta_{sf}-\delta }{\delta_{sf}-\delta_{s0}}(\delta >\delta_{s0}) \end{array}\right.$$
where $t_{n}$ and $t_{s}$ are the normal and tangential stress, respectively. Additionally, $t_{n0}$ and $t_{s0}$ are the maximum normal stress and maximum tangential stress, respectively. The relative displacement and the critical relative displacement when the normal stress reaches the maximum value are $\delta_{n0}$ and $\delta_{nf}$. $\delta_{s0}$ and $\delta_{sf}$ are the relative displacement and the critical relative displacement when the tangential stress reaches the maximum value. The material damage starts when the stress on the interface layer satisfies the quadratic nominal stress criterion in Equation (15).
(15)$${\left\{\frac{\langle t_{n}\rangle }{t_{n0}}\right\}}^{2}+{\left\{\frac{t_{s}}{t_{s0}}\right\}}^{2}+{\left\{\frac{t_{t}}{t_{t0}}\right\}}^{2}=1$$
where $t_{n}$ is the normal traction component, $t_{s}$ and $t_{t}$ are the tangential traction component, and $t_{n0}$, $t_{s0}$, and $t_{t0}$ are the critical stress at the beginning of the damage, respectively. The area enclosed by the traction-separation curve and axes in a horizontal and vertical direction is calculated by Equation (16). Additionally, $G_{nf}$ and $G_{sf}$ represent the fracture energy consumed per unit area for a normal and tangential crack propagation, respectively.
(16)$$G_{f}=\int_{0}^{\delta_{f}}t(\delta )d\delta =\frac{1}{2}t_{0}\delta_{f}$$

As illustrated in Figure 4, when the initial tensile stiffness $k_{n0}$ is determined, the traction-displacement curve can be determined by the fracture energy ($G_{nf}$), the maximum stress ($t_{n}$), and the critical relative displacement ($\delta_{nf}$). Additionally, a similar method can be applied to the tangential traction-displacement curve.

In order to obtain the material parameters of the bonding layer between the steel bar and concrete, the pull-out test of reinforcement was conducted with the YAW4306 electro-hydraulic servo testing machine. Before the test, a steel bar 400 mm in length and 12 mm in diameter embedded in the concrete block was applied. The bonding length of steel bar was set to 60 mm ($l_{bond}=5D_{s}$), as presented in Figure 5a. Additionally, the unbonded part of the steel bar was protected by PVC tubes with a length of 45 mm and a thickness of 2 mm. The setup of the pullout test is shown in Figure 5b. During the test process, the force and displacement of the loading point were recorded. Additionally, the average bond strength of the steel bar was calculated by the following formula:(17)$$\tau =\frac{F}{2\pi r_{s}l_{b}}$$
where τ is the bond strength between the reinforcement and concrete, $r_{s}$ is the radius of reinforcement, and $l_{b}$ is the bond length of the reinforcement.

For the purpose of verifying the effectiveness of the cohesive parameters obtained from pullout tests, the corresponding numerical model was established. The steel property was adopted for the space occupied by the steel bar, and the CDP model was adopted for the other parts. In addition, the cohesive elements with zero thickness were inserted between the concrete and steel bars, and the material parameters for the cohesive elements are shown in Table 3. Similar to the setup in the experiment, the ending of the unbonded steel bar below was fixed completely. Additionally, the unbonded steel bar above was set to move freely only in the vertical direction. Surface to surface contact was applied between the concrete and the fixed device. In the simulation, the fixed device was assumed to be a rigid body, and a constant loading rate v = 0.01 m/s was applied to the reference point on the fixed device.

During the numerical simulation, considering the elastic deformation of the steel bar generated during the loading process, the difference between the upper and lower ends of the steel bar is taken as the relative slip. Additionally, the reaction force obtained from the reference point is taken as the tensile force. According to Equation (17), the bond-slip curve is plotted. The comparison result is presented in Figure 6. As a whole, it can be seen that the numerical result is in a good agreement with the experimental result. With the movement of the rigid body in vertical direction, the stress of the interface layer begins to increase and the cohesive elements are stretched in a vertical direction. Subsequently, the cohesive elements closed to the loading point deforms seriously. After the bonding stress reaches the peak value, part of the cohesive elements whose stress reached the bond strength start to be deleted. During the softening stage of the bonding-slip curve, much more cohesive elements are deleted. When the bond between the reinforcement and the concrete is gradually destroyed, the reinforcement is pulled out from the concrete matrix.

### 3.2. Three-Point Bending Test Simulation

In order to simulate the dynamic failure process of notched reinforced concrete beams, the finite element model for the drop hammer impact system was established with the bond-slip constitutive model mentioned before, as shown in Figure 7. In the test, the drop hammer was divided into three parts: the hammerhead, force sensor, and additional weight. In the numerical simulation, a cylinder was established to simplify the drop hammer, which was set to be a rigid body, and the reference point could move only in a vertical direction. To simplify the calculation, an equivalent density (the actual mass of the falling hammer in the model divided by the volume of the model, about 67,570 kg/m^3^) was applied for the hammer. Two supports were assumed as rigid bodies, which could slide only in a horizontal direction. After analyzing the sensitivity of the mesh, 2 mm and 15 mm were adopted for the steel bar and concrete, respectively. Additionally, the three-dimensional 8-node reduced integral element (C3D8R) was adopted for the concrete and steel element types.

For brevity, a representative comparison is shown only for structural response when the impact velocity was 1.71 × 10^3^ mm/s. As presented in Figure 8, the experimental and numerical results are in a close agreement. It can be known that the vibration of the impact force decreased obviously in a numerical simulation. This results from the fact that the concrete was taken as a homogeneous and isotropic material and the separation between the concrete’s elements is not considered. As for the reaction force comparison, the numerical result is higher than the one in the experimental result due to the difference between the experiment and the simulation. In the numerical simulation, the sum of the contact forces between the supports and the bottom of the beam was taken as the reaction force. In the experiment, a steel support a 40 mm thickness, 400 mm in length, and a 150 mm width was installed between the force sensor and specimen to fix the force sensor on the rigid base. During the impact process, part of the impact energy was used for the accelerated movement and elastic deformation for the supports, which results in the reaction force measured from the test being small than the numerical result. As shown in Figure 8c, the initial increase stage of the mid-span deflection obtained from the experiment and simulation presents a great agreement. Interestingly, it is found that the numerical result of the maximum mid-span deflection is less than the experimental result, which may result from the fact that the energy stored in the matrix was released as the crack propagated. In the numerical simulation, only the failure of the bonding layer between the concrete and reinforcement and tension/compression damage of the concrete are simulated; the separation between the concrete elements is not considered.

The reinforcement ratio is an important material parameter to describe reinforced concrete structures. However, under the same reinforcement ratio, significant differences are found in the bonding area of concrete specimens with steel bars of different diameters. In the experiment, the diameter of the steel bar was 12 mm, and the reinforcement ratio was 0.5%. By keeping the reinforcement ratio, depth, and width of the notch, the distance between the reinforcement and the bottom of the specimen unchanged, and by inserting steel bars with different diameters into the concrete matrix, the effect of the reinforcement configuration on the dynamic failure process of the reinforced notched beams can be studied. As illustrated in Figure 9, the radius of the steel bars was set to be 6 mm, 4.24 mm, and 3 mm, respectively. The notch depth and width for all specimens are 45 mm and 4 mm, respectively. Additionally, the distance from the steel bar to the bottom of the beam for all models is 30 mm, as same as the specimens in tests. Three kinds of reinforced concrete notched beams are marked as d1, d2, and d4, respectively. Additionally, the bonding area can be calculated by the following formula:(18)$$S_{b}=2\pi r_{s}l_{b}$$
where $r_{s}$ is the reinforcement radius and $l_{b}$ is the bonding length. In the numerical model, the reinforcement is lower than the notch root. The width of the notch is 4 mm, and the length of the specimen is 800 mm, so the bond length of the three specimens is 796 mm. According to the formula mentioned before, the bonding areas of the three numerical models are 0.03 m^2^, 0.04 m^2^, and 0.06 m^2^, respectively.

## 4. Experimental Results

### 4.1. Failure Process

Considering the low reinforcement ratio and low impact velocity, the crack at the mid-span of the beam initiates from the notch root and then propagates upwardly to the loading point. Thus, a spray-painted speckle pattern is applied onto the mid-span surface of the specimens to monitor the crack propagation process. Additionally, then Digital Image Correction (DIC) technology is conducted to measure the displacement field. Subsequently, the strain field is given from the displacement field with the Euler–Lagrange algorithm in MATCH-ID 2021.2, MATCH-ID NV (Ghent, Belgium). Finally, the crack path is obtained from the change in the strain field in the horizontal direction (*Exx*) under different impact loading, as presented in Figure 10. The blue dotted circle and white dotted line represent the crack tip and crack propagation path, respectively.

Under different impact velocities, the crack initiates from the notch root and then propagates. As the unloading of the hammer begins, the elastic deformation of the beam starts to recover, which results in the occurrence of the closure phenomenon. With the increase in the impact energy, the plastic deformation of the steel bar generates. It is apparent that the crack length increased as the impact velocity increased. When the loading time was 2 ms, it is found that the crack length increases significantly as the impact velocity was enhanced. This phenomenon results from the enhancement of the impact energy. Similarly, the crack opening displacement increased greatly when the loading time is 4 ms. Differing from the failure pattern, the impact velocity was 9.89 × 10^2^ mm/s and 1.71 × 10^3^ mm/s; crushing failure was observed at the upper surface closed to the loading point for an impact velocity of 2.21 × 10^3^ mm/s. Additionally, the crush crack meets the crack initiated from the notch root gradually.

### 4.2. Structural Response

Three repeated tests were carried out for each specimen, and only one of the repeated tests was selected to make a comparative analysis due to the excellent repeatability. As described in Figure 11, the time curves for the impact force, reaction force, and steel strain at the mid-point are plotted. Under different impact velocities, the impact force increased rapidly at the beginning of the loading, while an increase in the reaction force was hard to find. This results from the impact force being almost completely balanced by the inertial force at the initial stage.

When the impact velocity is 0.989 m/s, the platform of the impact force appears after reaching the peak. Meanwhile, the reaction force increases to the maximum value. Compared to the impact force, the reaction force time curves are exhibited as half sine waves. When the impact velocity is higher than 0.989 m/s, the vibration of the impact force aggravates significantly, while the reaction force is similar to the one in 0.989 m/s. Because the vibration for impact force increased due to the inertial effect and increase in the impact velocity, the reaction force with only one peak load better reflects the impact resistance of the component than the impact force [21,22].

Under a low impact velocity (0.989 m/s), the maximum value of the steel strain is about 1000 με, which indicates that only elastic deformation occurs during the impact process. It is worth noting that the maximum value for the steel strain reaches 3000 με with the increase in the impact velocity. Considering that the inelastic strain of steel is about 2500 με, it is obvious that the steel at the mid-span reaches the plastic stage. Subsequently, the decrease in the steel strain indicates the recovery of the elastic deformation for the steel bar.

Apart from the dynamic bearing capacity, the absorbed energy is another important parameter in evaluating the structural safety. The area enclosed by the reaction force-mid-span deflection curve can be taken as the energy absorbed in the failure process of the beam [17]. The reaction force versus the mid-span deflection curves under three impact velocities are illustrated in Figure 12. It is apparent that both the maximum deflection and residual deflection increase with the growth of the impact velocity. This is related to the recovery of the elastic deformation weakening with the increase in the impact energy. After integrating the curves from Figure 12, the energy absorbed under three impact velocities is 8.45 J, 37.01 J, and 88.33 J, respectively. It can be seen from Figure 10, only when the impact velocity is 2.21 m/s, that the crack extends to the loading point. Although the steel bar is not broken, the concrete is destroyed completely. Thus, the energy absorbed during the failure process is higher when the impact velocity is 2.21 m/s.

## 5. Numerical Results

### 5.1. Failure Pattern

Based on the numerical model mentioned before, three impact velocities for 0.5 m/s, 1.5 m/s, and 2.5 m/s are assigned to the hammer reference points in a vertical direction to study the failure pattern for specimens with different amounts of steel bars. The dynamic failure process is shown in Figure 13. When the impact velocity is 0.5 m/s, it is apparent that only the concrete at the mid-span is damaged, and the bending failure dominates the structural response for specimens with different bonding areas. As the impact velocity increased to 1.5 m/s, except for the crack at the mid-span, some bending cracks at both sides are found, and the concrete damage weakens as there is an increase in the bonding area. This phenomenon may result from the increase in the bonding area promoting the stress propagating widely in concrete. When the impact velocity is 2.5 m/s, a symmetrical oblique crack occurs in d1 and d2, as the drop hammer starts to contact with the upper surface of the beams. As for the concrete beam with four steel bars, some shear cracks are formed from the support to the loading point as there is an increase in the applied load. It can be seen that the increase in bonding area leads to the enhancement of the damage of the concrete near the steel bars. In conclusion, the effect of the bonding area on the failure pattern can be ignored under a low impact velocity. As the bonding area increased, the failure pattern changes from bending failure to shear failure.

### 5.2. Structural Response

The reaction force and mid-span deflection from numerical results are obtained to evaluate the absorbed energy. Based on the method used in the previous tests, the energy consumed by the specimens during the impact process is calculated by integrating the reaction force-mid-span deflection curves. The impact energy during the impact process is the kinetic energy of the hammer, which can be calculated according to $E_{i}=m\nu^{2}/2$. The impact energy corresponding to the three impact velocities are: 13.25 J, 119.25 J, and 331.25 J, respectively. Meanwhile, the static three bending tests for different specimens are conducted to obtain the static bearing capacity, whose loading rate is 1 mm/s. The static bearing capacities of the three specimens are 28.38 kN, 31.82 kN, and 40.14 kN, respectively. The numerical results are shown in Table 4.

According to the static simulation results, the static bearing capacity of the specimens increases with the growth in the bonding area. To evaluate the effect of the bonding area on the dynamic bearing capacity, the ratio of the maximum reaction force ($R_{\max}$) to the static bearing capacity ($P_{u}$) is taken as the dynamic response ratio. As shown in Figure 14, the dynamic response ratio for all the specimens increases linearly with the growth of the impact velocity. Additionally, the fitted line can be described as follows:(19)$$\frac{R_{\max}}{P_{u}}=0.41\nu +0.46$$
where ν is the impact velocity. It can be seen that the effect of the bonding area on the dynamic response ratio presents different trends under three impact velocities. Under a low impact velocity (1.5 m/s), the influence of the bonding area can be ignored, which is consistent with the failure pattern. As the impact velocity increases to 1.5 m/s, the dynamic response ratio decreases with the growth of the bonding area. However, the opposite trend is found as the impact velocity increased to 2.5 m/s. The reason for this phenomenon is that the shear failure pattern occurs when v = 2.5 m/s, which leads to the increase in the reaction force being higher than the static bearing capacity.

In the numerical simulation, the maximum mid-span deflections of concrete specimens with different reinforcement conditions are extracted, as presented in Figure 15. It can be seen that the maximum mid-span deflections of different specimens increase linearly with the increase in the impact velocity. After fitting it, the empirical formula is given:(20)$$\delta_{\max}=1.4\nu +0.67$$

The effect of the bonding area on the deflection can be ignored when the impact velocity is 0.5 m/s and 1.5 m/s. When the impact velocity is 2.5 m/s, the maximum mid-span deflection decreases more obviously as the bonding area increases. Under the same impact energy, the steel bars with a small radius reach the plastic stage easily, which leads to the deformation resistance for the increases in the reinforced concrete beam.

As presented in Figure 16, the ratio of the absorbed energy to the input energy decreases with the growth of the impact velocity. This results from the fact that much more energy is consumed by the inertial force as the impact velocity increased. It is apparent that the growth of the bonding area results in the ratio tending to be stable, which is meaningful to predict the energy absorbed under a certain impact velocity.

### 5.3. Energy Dissipation for Components

For the purpose of investigating the dynamic failure process, the energy dissipation for concrete, steel bars, and the bonding layer are analyzed. The damage energy dissipation of concrete under different velocities are compared, as shown in Figure 17. It is apparent that the effect of the bonding area on the initial growth rate of concrete damage energy dissipation under three impact velocities can be ignored. This results from the fact that the concrete response dominants when the hammer is in contact with the upper surface of the concrete beam [16]. When the impact velocity is less than 2.5 m/s, the damage energy dissipation for concrete increases with the growth of the bonding area, whereas the opposite trend is observed when v = 2.5 m/s. This results from the failure pattern changing from the bending failure to the shear failure at 2.5 m/s and much more energy is consumed for the plastic deformation of steel bars. It can be seen that the damage for concrete mainly occurs at the mid-span when a bending failure occurs, and the damage for concrete near the steel bars enhances when the shear cracks generate. In conclusion, the damage energy dissipation for concrete increases gradually with the bonding area when the bending failure occurs. Under shear failure, the concrete damage energy dissipation reduces.

In the numerical model, the bonding layer between the steel bar and the concrete is modeled with cohesive elements of a zero thickness, whose plastic deformation is not found. Therefore, only the plastic energy dissipation for steel bars and concrete are analyzed, as shown in Figure 18. When the impact velocity is 0.5 m/s and 1.5 m/s, the plastic energy dissipation of the steel bars in all the specimens is zero, which indicates that only elastic deformation generates. Additionally, it is obvious that the greater the bonding area is, the more plastic energy dissipation there is for concrete. When the impact velocity is 2.5 m/s, the plastic energy dissipation of steel bars in all of the specimens increase significantly and the trigger time decreases with the growth of the bonding area. This may result from the fact that the smaller the reinforcement diameter is, the easier it is for the plastic deformation to occur. Additionally, the plastic energy dissipation for concrete decreases along with the growth of the bonding area, whereas the plastic energy dissipation for steel bars presents the opposite trend. Therefore, the deformation of the steel bar gradually replaces concrete failure and plays a dominant role in the failure process.

The internal energy dissipation for each component is compared and analyzed, as shown in Figure 19. As the impact velocity increased, the internal energy dissipation for all components increased significantly. When the impact velocity was 0.5 m/s, the internal energy dissipation for the concrete increased with the growth of the bonding area, whereas that for the bonding layer presents the opposite trend. This indicates that the increase in the bonding area leads to the easier transportation of stress from steel bars to concrete, and then the deformation area of the concrete enhances. Additionally, the influence from the bonding area on the internal energy dissipation for steel bars can be ignored. As the impact velocity increases to 1.5 m/s, the internal energy dissipation for each component is similar to that at 0.5 m/s due to the similar failure pattern. When the impact speed is 2.5 m/s, the internal energy dissipation for the concrete and bonding layer decreases gradually with the increase in the bonding area. Nevertheless, the obvious growth trend of the internal energy dissipation for steel bars is found. The reason for this phenomenon is that the steel bars reach the plastic stage easily with the increase in the bonding area. In conclusion, when the reinforcement ratio remains constant, the increase in the bonding area leads to the enlargement for a mid-span deflection. When the shear failure occurs in reinforced concrete beams, the internal energy dissipation for steel bars enhances along with the growth of the bonding area due to the increase in the plastic deformation.

Although the specimens with different bonding areas have the same failure pattern when the impact velocity is less than 2.5 m/s, a difference in the stress distribution is found, as shown in Figure 20. When v = 0.5 m/s, at the beginning of the impact process, the stress of the bonding layer at the mid-span increases rapidly to the maximum value, and the stress gradually weakens along the longitudinal direction. It is apparent that the maximum stress decreases with the growth of the bonding area. Thus, the growth of the bonding area effectively relieves the stress concentration at the mid-span, which avoids the bonding layer being destroyed. When the impact velocity is 1.5 m/s, the impact process can be divided into two stages: the loading stage and the unloading stage. In the initial stage, the stress distribution of the bonding layer is similar to the one at 0.5 m/s. With the rebound of the hammer, the recovery of the elastic deformation for the bonding layer leads to the stress of the bonding layer at the mid-span and the supports are higher than that between them. It is apparent that the maximum stress of the bonding layer decreases with the bonding area. As the impact velocity increased to 2.5 m/s, the stress distributions of the bonding layer in all of the specimens are similar to that at 1.5 m/s, and smaller stress values and wider stress distributions are presented.

In conclusion, when the reinforcement ratio is a constant value, the increase in the bonding area leads to the decrease in the maximum stress of the bonding layer, which reduces the failure of the bonding layer and protects the integrity of the specimen.

## 6. Conclusions

For the purpose of investigating the effect of the bonding area on the dynamic fracture behavior for notched reinforced concrete beams with the same reinforcement ratio, dynamic three-point bending tests and a numerical simulation are carried out. Additionally, the effects of the bonding area on the energy dissipation for each component are discussed. The main findings are summarized as follows:When the reinforcement ratio remains constant, with the growth of the impact velocity and bonding area, the failure pattern for the notched reinforced concrete beam changes from bending failure into shear failure.As the bonding area increases, the damage energy dissipation for concrete decreases when shear failure occurs, which is opposite to that in the bending failure pattern.The larger the bonding area is, the higher the plastic energy dissipation for steel bars is and the smaller the mid-span deflection is. This indicates that the impact resistance enhanced with the growth of the bonding area.Under different impact velocities, the internal energy dissipation of the bonding layer decreases with the growth of the bonding area. Therefore, the increase in the bonding area can protect the bonding effect between steel bars and concrete.

## Figures and Tables

**Figure 1 materials-16-00507-f001:**
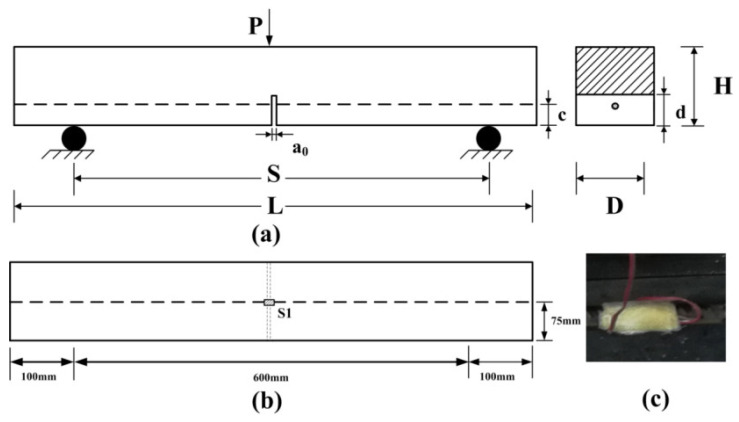
The setup of specimen: the schematic diagram of the specimens (**a**); the location (**b**) and photo (**c**) of strain gauge on the steel bar.

**Figure 2 materials-16-00507-f002:**
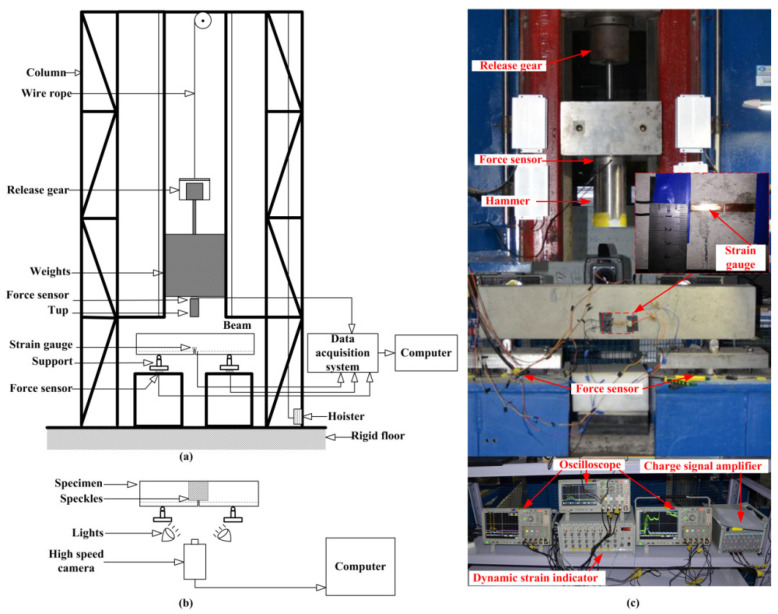
Schematic diagram of drop-hammer impact machine (**a**) and DIC equipment (**b**); photo of impact experiment setup (**c**).

**Figure 3 materials-16-00507-f003:**
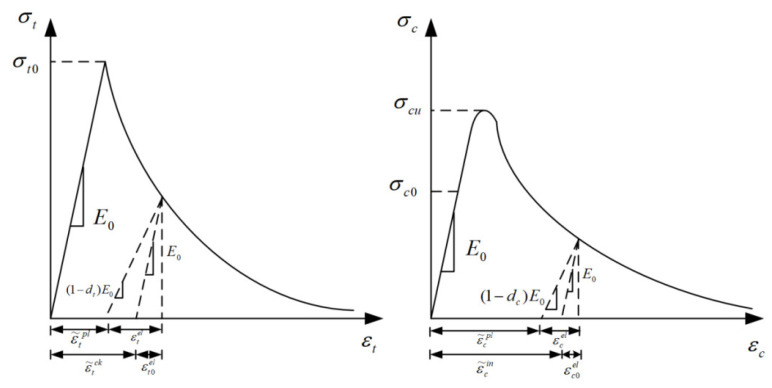
Uniaxial tensile/compressive stress–strain curve of concrete.

**Figure 4 materials-16-00507-f004:**
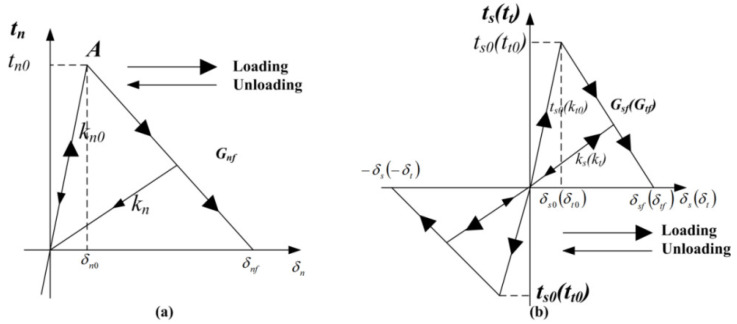
The bilinear constitutive for bonding layer: (**a**) in normal direction; (**b**) in tangential direction.

**Figure 5 materials-16-00507-f005:**
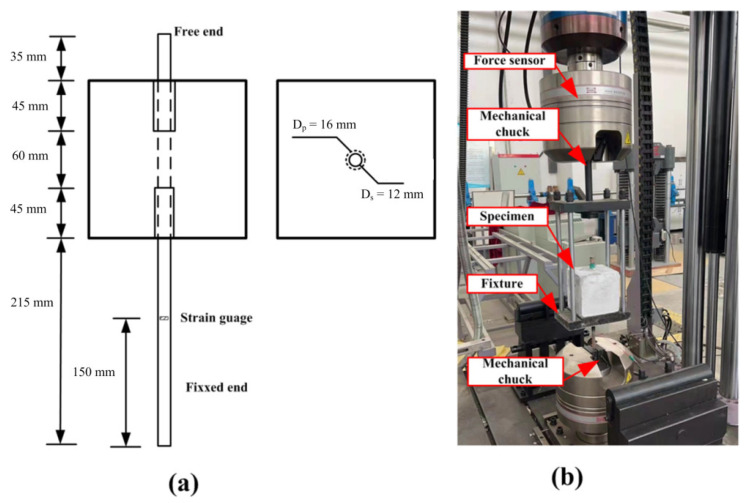
Pullout test: (**a**) design of specimen; and (**b**) setup of the pull-out experiment.

**Figure 6 materials-16-00507-f006:**
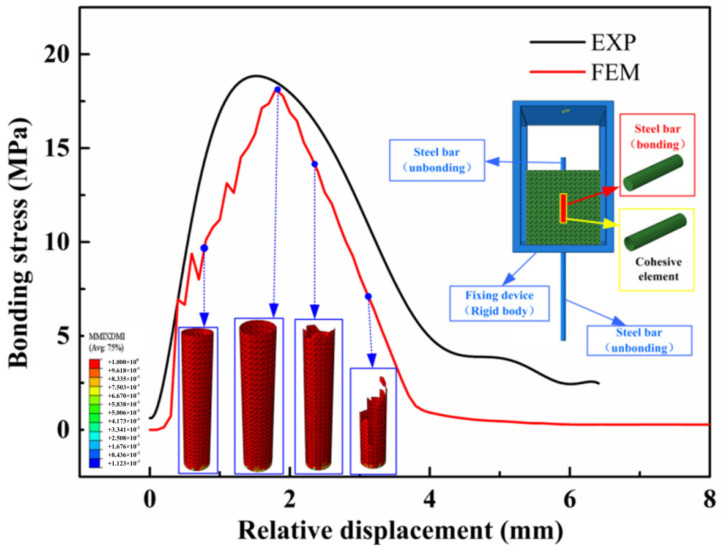
Comparison between numerical and experimental results.

**Figure 7 materials-16-00507-f007:**
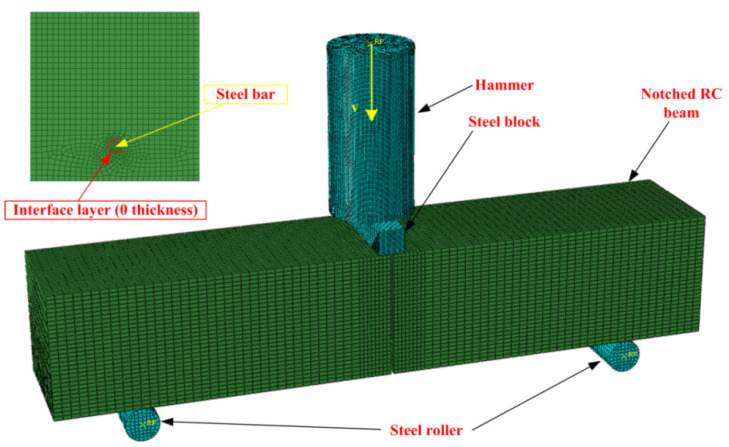
The numerical model for drop hammer impact system.

**Figure 8 materials-16-00507-f008:**
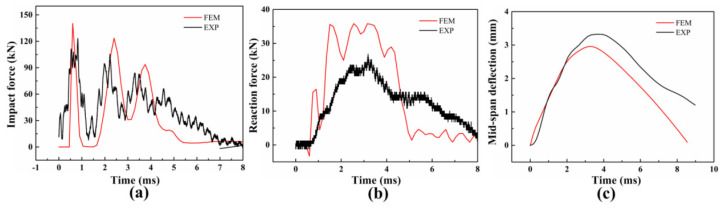
The comparison of (**a**) impact force; (**b**) reaction force; and (**c**) mid−span deflection time curves between experimental and numerical results.

**Figure 9 materials-16-00507-f009:**
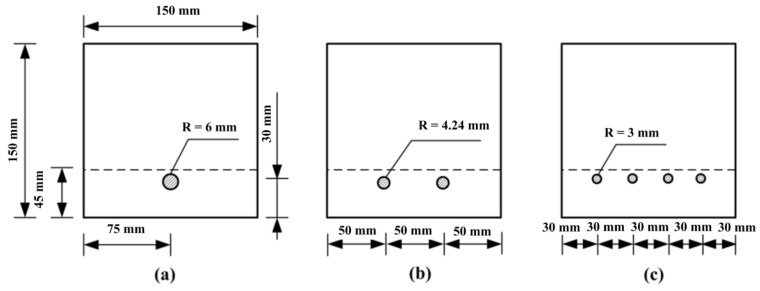
The arrangement of steel bars in the numerical models with: (**a**) one steel bar; (**b**) two steel bars and (**c**) four steel bars, respectively.

**Figure 10 materials-16-00507-f010:**
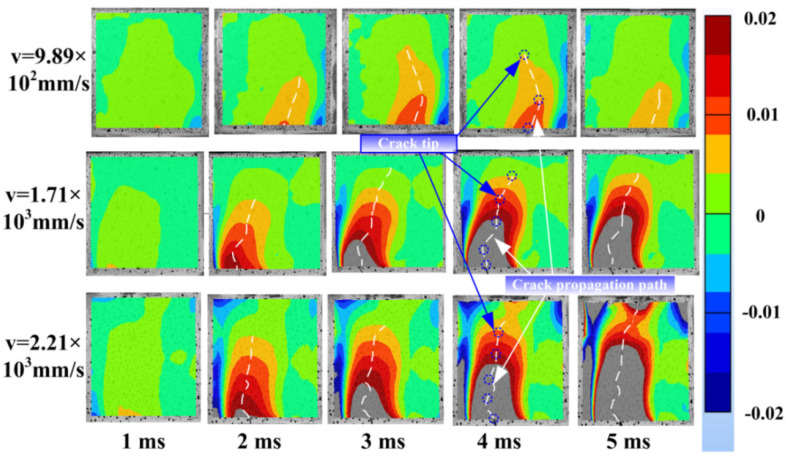
The strain filed in horizontal direction.

**Figure 11 materials-16-00507-f011:**
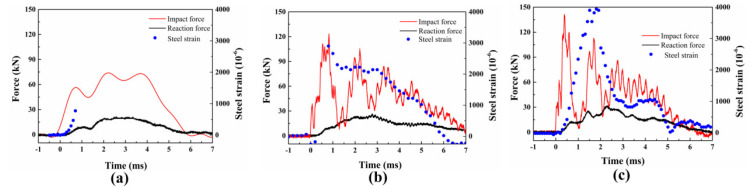
The experimental results under (**a**) 0.989 m/s; (**b**) 1.71 m/s; and (**c**) 2.21 m/s.

**Figure 12 materials-16-00507-f012:**
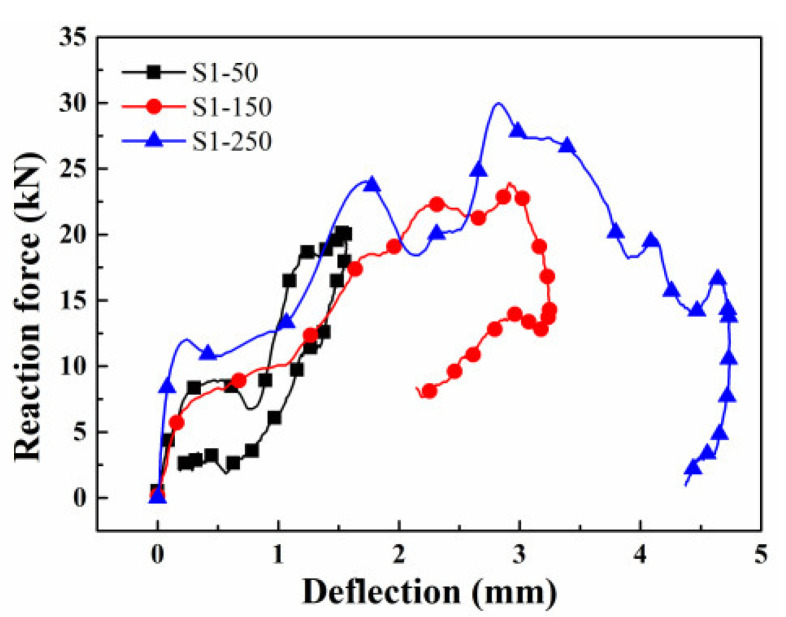
The reaction force versus mid-span deflection.

**Figure 13 materials-16-00507-f013:**
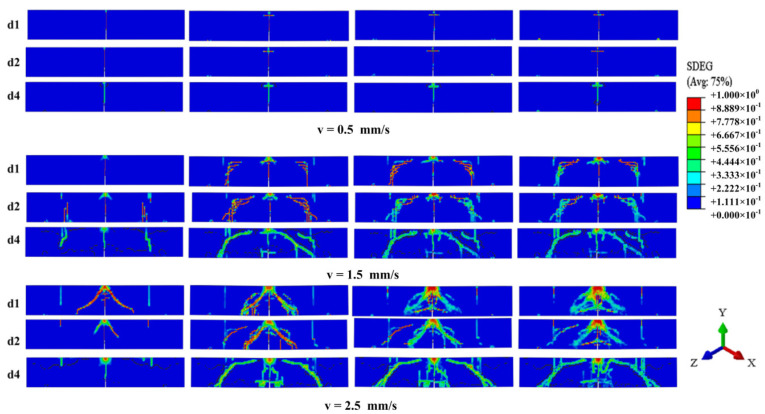
The dynamic failure process of reinforced concrete beams with different bonding area.

**Figure 14 materials-16-00507-f014:**
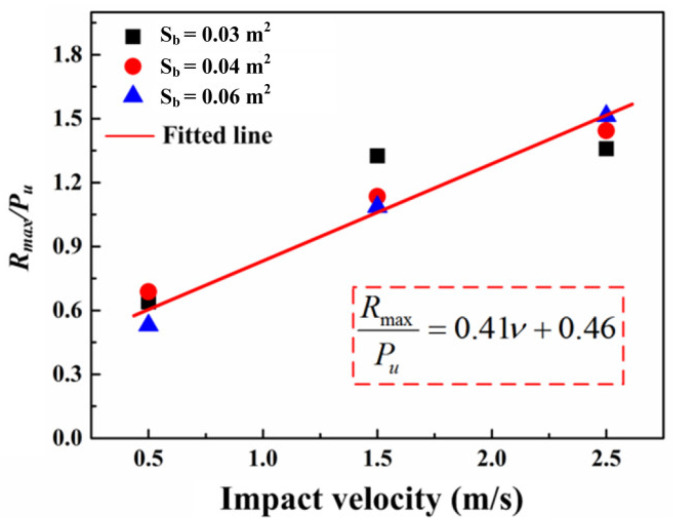
The relation between dynamic response and impact velocity.

**Figure 15 materials-16-00507-f015:**
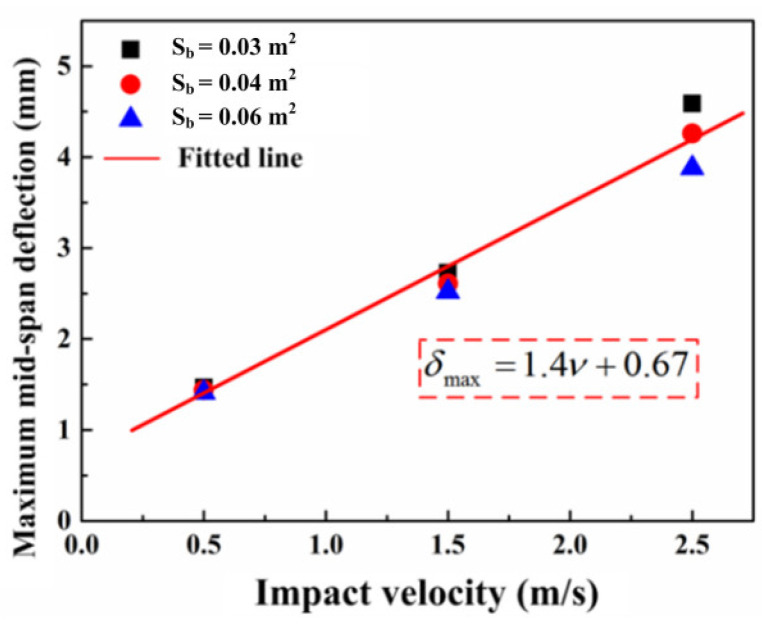
The relation between maximum mid-span deflection and (**a**) reaction force or (**b**) the number of steel bars.

**Figure 16 materials-16-00507-f016:**
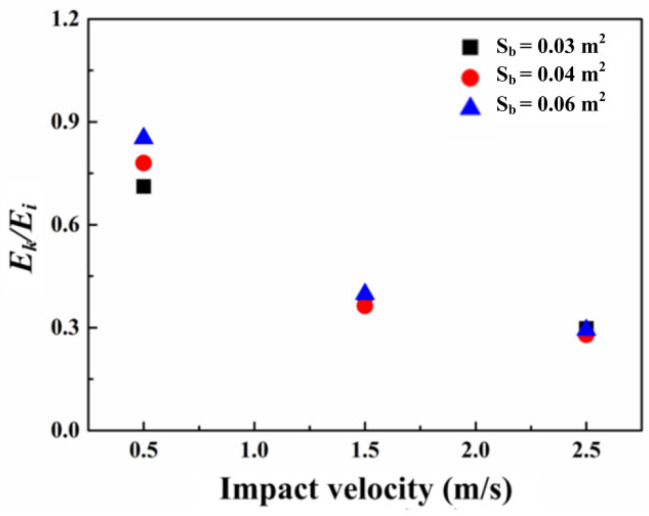
The influence of impact velocity on absorbed energy during the failure process.

**Figure 17 materials-16-00507-f017:**
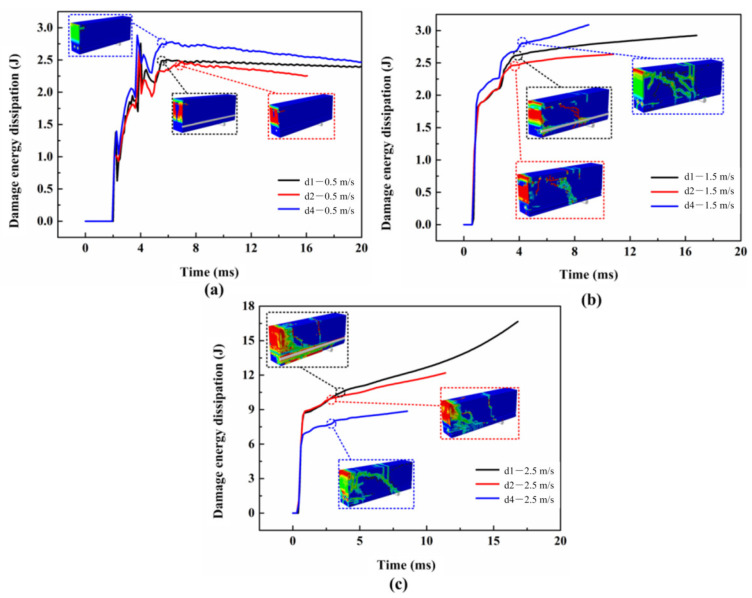
The damage energy of concrete when impact velocities are (**a**) 0.5 m/s; (**b**) 1.5 m/s; and (**c**) 2.5 m/s.

**Figure 18 materials-16-00507-f018:**
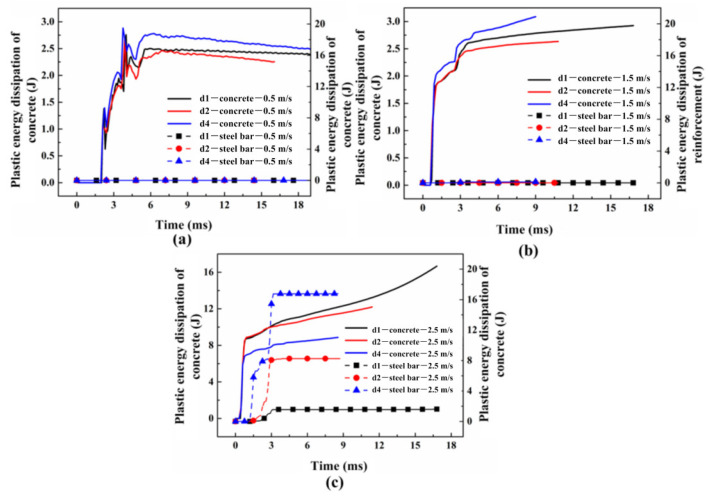
The plastic dissipation for concrete and steel bars when impact velocity are (**a**) 0.5 m/s; (**b**) 1.5 m/s; and (**c**) 2.5 m/s.

**Figure 19 materials-16-00507-f019:**
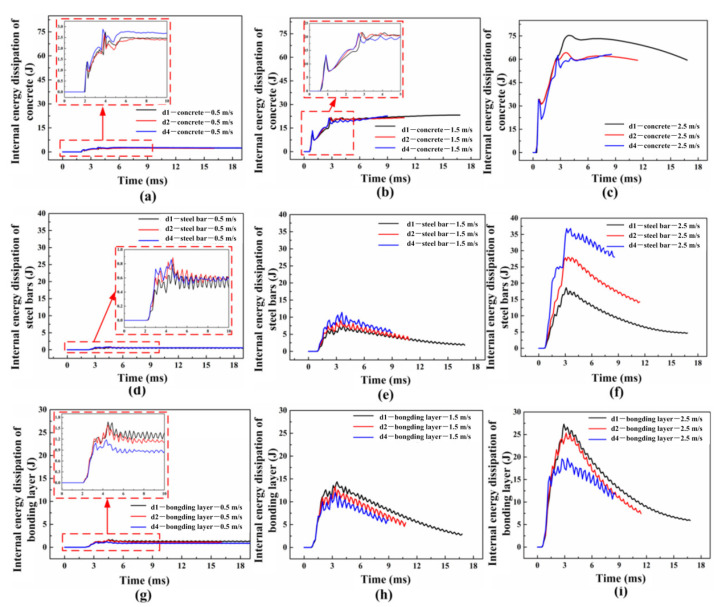
The internal energy dissipation for each of the components under three impact velocities.

**Figure 20 materials-16-00507-f020:**
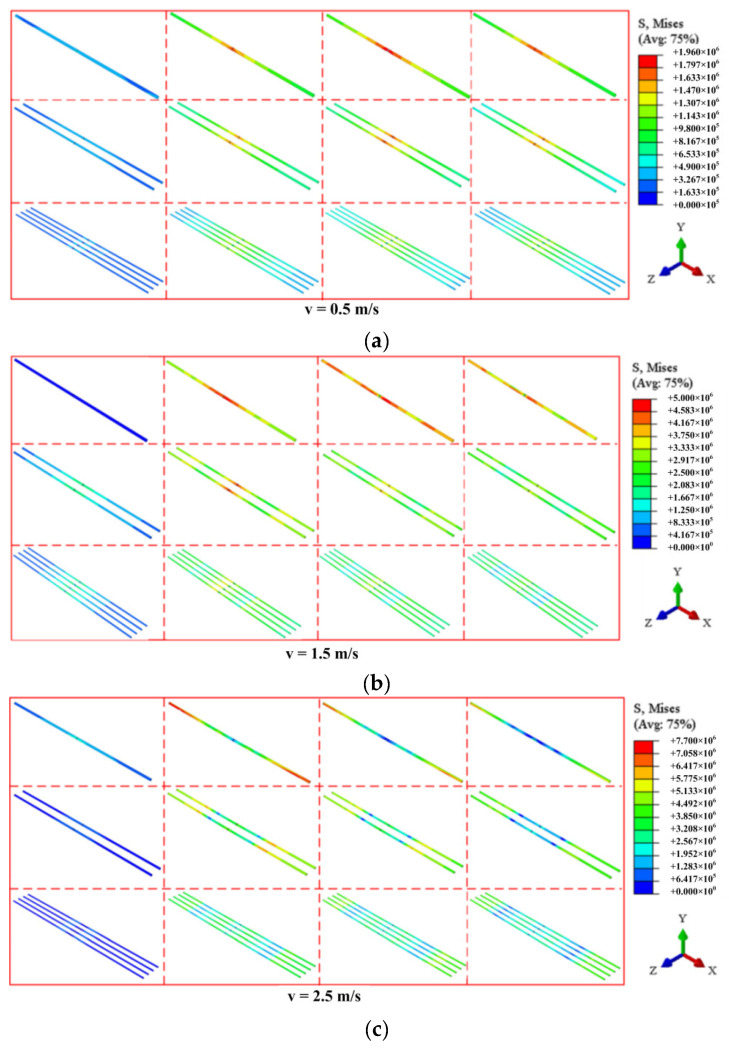
The stress trend of bonding layer when impact velocity is (**a**) 0.5 m/s; (**b**) 1.5 m/s and (**c**) 2.5 m/s.

**Table 1 materials-16-00507-t001:** Parameters of concrete [34].

Material	Dilation Angle (°)	Eccentricity	$f_{b0}/f_{c0}$	K	Viscosity Parameter	Density (kg/m^3^)	Elastic Modulus (GPa)	Poisson’s Ratio
Concrete	30	0.1	1.16	0.6667	0.0005	2400	26.48	0.167

**Table 2 materials-16-00507-t002:** Parameter of steel model [43].

Steel Grades	Diameter (mm)	Density (kg/m^3^)	Elastic Module (GPa)	Poisson’s Ratio	Yield Strength (MPa)	Ultimate Strength (MPa)
HRB335	12	7850	200	0.3	438	687

**Table 3 materials-16-00507-t003:** Material properties.

Component	Parameter	Value
Bonding layer	Stiffness (E_1_ = G_1_ = G_2_)	1 × 10^9^ Pa/m
	Nominal stress normal-only mode	2 × 10^7^ Pa
	Nominal stress first direction	2 × 10^7^ Pa
	Nominal stress second direction	2 × 10^7^ Pa
	Displacement at failure	2 × 10^−3^ m

**Table 4 materials-16-00507-t004:** The numerical results of notched beams with different setups of steel bars.

Specimens	Static Bearing Capacity $P_{u}$ (kN)	Maximum Reaction Force $R_{\max}$(kN)	$R_{\max}/P_{u}$	Impact Velocity (m/s)	Impact Energy $E_{i}$ (J)	Maximum Mid-Span Deflection (mm)	Absorbed Energy $E_{k}$ (J)	$E_{k}/E_{i}$
d1	28.38	18.15	0.64	0.5	13.25	1.47	9.43	0.71
	28.38	37.59	1.32	1.5	119.25	2.73	44.16	0.37
	28.38	38.57	1.36	2.5	331.25	4.59	98.56	0.30
d2	31.82	21.86	0.69	0.5	13.25	1.44	10.33	0.78
	31.82	36.10	1.13	1.5	119.25	2.61	43.3	0.36
	31.82	45.92	1.44	2.5	331.25	4.26	92.49	0.28
d4	40.14	21.26	0.53	0.5	13.25	1.41	11.29	0.85
	40.14	43.6	1.09	1.5	119.25	2.52	47.41	0.40
	40.14	60.72	1.51	2.5	331.25	3.88	97.04	0.29

## Data Availability

Not applicable.

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
