# Peer review of "Influence of Bonding Area on Dynamic Failure Behavior of Notched Reinforced Concrete Beams"

_materials, 2023, doi:10.3390/ma16020507_

Round 1
Reviewer 1 Report
|
Dear Editor-in-Chief: Journal Materials (ISSN 1996-1944)
Manuscript ID: materials-2023668
|
This manuscript presented the Dynamic failure behavior of notched reinforced concrete beam with same reinforcement ratio but different bonding area. The paper is worth publishing however, the following concerns should be addressed in revising the manuscript:
· Abstract
1-What new information does the study provide? What sets this manuscript apart from others that present similar findings?
2. Highlight pivotal findings that prove the study's hypotheses and/or predictions.
· Keywords
3- This keyword is too long "notched reinforced concrete beam"
· Introduction
4- It is preferable to avoid the word “Many researchers studied” mentioning the role of each reference.
5- It is not preferable to develop programs collectively "[9-20]". It is better to mention the name of each researcher separately.
6. The scientific basis of the research should be covered in a separate section, if possible.
7. It is preferable to be clearer about what is novel about this study in terms of the methodologies used or the way it was created in collaboration, etc.
· Experimental program
8- The applicable standards for each step should be mentioned.
· Results and discussion
9- It is preferable to link the results with each other and support them with previous research.
· Conclusion
10- It is preferable to show the results achieved and related to the goals of the study.
11-It is preferable to display the values of the most important results.
thank you
Reviewer 2 Report
Dear authors of the manuscript entitled “Dynamic failure behavior of notched reinforced concrete beam with same reinforcement ratio but different bonding area”,
I have carefully reviewed your manuscript. The topic seems fit with the scope of the journal. However, you are required to address the following issues before considering the manuscript for publication:
1- Please re-write the title of research according to journal style.
2- There are some typing and grammar mistakes. Therefore, I suggest reviewing the entire manuscript by a professional editing service
3- The research problem is not clearly mentioned at the beginning of the abstract section. Therefore, please reconstruct the abstract accordingly.
4- More recent literature should be included in the introduction to add power to the research. Therefore, please consider revision.
5- Some symbols need to be defined in the abbreviations paragraph. Please consider revision
6- References are not all in the same format. Therefore, please use proper tool for citation (e.g. Endnote or Mendeley) and follow the journal style.
7- Please download the citation from original source (e.g science direct, springer, …etc.). not simply use google scholar. Moreover, it is strongly encouraged to include the DOI of all references.
8- Finally, I have not seen any connection between your research and articles already published in this journal. Please consider this matter.
Therefore, the reviewer believes that the manuscript is not suitable for publication in its present form due to the above reasons.
Round 2
Reviewer 1 Report
Dear,
The paper in its current state is significantly improved.
The authors have responded to all comments, and the paper can be published
Reviewer 2 Report
Dear authors of the manuscript entitled “Dynamic failure behavior of notched reinforced concrete beam with same reinforcement ratio but different bonding area”,
I have carefully reviewed your revised manuscript and have concluded the following remarks:
1- The manuscript is generally well-revised with a comprehensive introduction, adequate methodology, and an intense discussion of the results including tables, figures, diagrams, …etc.
2- Author responses to reviewers’ comments are well addressed.